# Accuracy of digital chest x-ray analysis with artificial intelligence software as a triage and screening tool in hospitalized patients being evaluated for tuberculosis in Lima, Peru

Amanda M. Biewer[1], Christine Tzelios[2], Karen Tintaya[3], Betsabe Roman[3], Shelley Hurwitz[2], Courtney M. Yuen[4], Carole D. Mitnick[4], Edward Nardell[4], Leonid Lecca[3], Dylan B. Tierney[4,5], Ruvandhi R. Nathavitharana[1] *

1 Beth Israel Deaconess Medical Center and Harvard Medical School, Boston, Massachusetts, United States of America, 2 Harvard Medical School, Boston, Massachusetts, United States of America, 3 Socios en Salud, Lima, Peru, 4 Brigham and Women's Hospital and Harvard Medical School, Boston, Massachusetts, United States of America, 5 Massachusetts Department of Public Health, Boston, Massachusetts, United States of America

☯ These authors contributed equally to this work.
* rnathavi@bidmc.harvard.edu

## Abstract

Tuberculosis (TB) transmission in healthcare facilities is common in high-incidence countries. Yet, the optimal approach for identifying inpatients who may have TB is unclear. We evaluated the diagnostic accuracy of qXR (Qure.ai, India) computer-aided detection (CAD) software versions 3.0 and 4.0 (v3 and v4) as a triage and screening tool within the FAST (Find cases Actively, Separate safely, and Treat effectively) transmission control strategy. We prospectively enrolled two cohorts of patients admitted to a tertiary hospital in Lima, Peru: one group had cough or TB risk factors (triage) and the other did not report cough or TB risk factors (screening). We evaluated the sensitivity and specificity of qXR for the diagnosis of pulmonary TB using culture and Xpert as primary and secondary reference standards, including stratified analyses based on risk factors. In the triage cohort (n = 387), qXR v4 sensitivity was 0.91 (59/65, 95% CI 0.81–0.97) and specificity was 0.32 (103/322, 95% CI 0.27–0.37) using culture as reference standard. There was no difference in the area under the receiver-operating-characteristic curve (AUC) between qXR v3 and qXR v4 with either a culture or Xpert reference standard. In the screening cohort (n = 191), only one patient had a positive Xpert result, but specificity in this cohort was high (>90%). A high prevalence of radiographic lung abnormalities, most notably opacities (81%), consolidation (62%), or nodules (58%), was detected by qXR on digital CXR images from the triage cohort. qXR had high sensitivity but low specificity as a triage in hospitalized patients with cough or TB risk factors. Screening patients without cough or risk factors in this setting had a low diagnostic yield. These findings further support the need for population and setting-specific thresholds for CAD programs.

**Data Availability Statement:** Data files uploaded to Harvard Dataverse. https://dataverse.harvard.edu/dataset.xhtml?persistentId=doi:10.7910/DVN/EOWYEQ.

**Funding:** This work was supported by the National Institutes of Health (NIAID R01 AI112748 to EN and DT, and NIAID K23 AI132648-05 to RRN) and American Society of Tropical Medicine and Hygiene (Burroughs Wellcome Fellowship to RRN). The funders had no role in study design, data collection, data analysis, data interpretation, writing of the report, or in the decision to submit for publication. The content is solely the responsibility of the authors and does not necessarily represent the views of the funders.

**Competing interests:** The authors declare that no competing interests exist.

## Introduction

Diagnosis remains the largest gap in the tuberculosis (TB) cascade of care. In 2021, of the 10.6 million people estimated to become sick due to TB, only 6.4 million were diagnosed and notified to national notification systems [1]. Efforts to increase and accelerate diagnoses are critical to prevent severe disease, avert TB deaths, and halt ongoing transmission [2]. Healthcare facilities are known hotspots for TB transmission in high-incidence settings [3–7]. Globally, the rate of TB disease among healthcare workers is estimated to be at least double that of the general adult population, suggesting significant transmission in health facilities [8, 9]. The FAST (Find cases Actively, Separate safely, and Treat effectively) strategy was developed to reduce TB transmission in healthcare settings, based on the principle that most transmission occurs from patients with unsuspected and thus undiagnosed TB, including drug-resistant strains [10]. FAST relies on identifying potentially infectious patients, typically with cough screening, followed by rapid sputum-based molecular tests that include first line resistance testing to enable prompt initiation of effective treatment [7, 10]. FAST has been implemented in a variety of settings, including Peru, Bangladesh, Russia, and Vietnam [11–14]. Given the slow scale up of rapid molecular tests [1], due to barriers such as cost, optimizing screening approaches for the FAST strategy is critical for its implementation success.

Triage is the process of making clinical decisions based on symptoms, signs, risk factors, or test results [15]. Rapid and accurate triage tests play an important role in identifying patients requiring further diagnostic evaluation among those with symptoms or risk factors for disease [16]. Screening similarly involves non-diagnostic testing to distinguish between people who likely have the disease from those who are unlikely to have the disease, typically in a population who do not have symptoms [15]. There is a long history of using chest radiography (CXR) to screen for pulmonary TB, but its utility in high TB incidence settings has been limited by the scarcity of skilled radiologists to interpret images [17]. The advent of digital radiography coupled with computer aided detection (CAD) software eliminates this potential barrier, making it more feasible to implement CXR for triage or screening in resource limited settings. CAD uses artificial intelligence algorithms to analyze radiographs for abnormalities consistent with TB. CAD is now recommended by the World Health Organization (WHO) as an alternative to human readers[15]. Nonetheless, while CAD sensitivity for both triage and screening is typically >90%, CAD specificity varies widely, from 23%–66% for screening[15,18,19] and 25%–79% for triage[18,20] when compared to a microbiological reference standard.

Questions remain regarding the optimal approach for using CAD to identify potentially infectious people with TB, particularly in hospital settings. A retrospective case-control study evaluating CAD in patients presenting with respiratory symptoms to a tertiary care hospital in India demonstrated moderate sensitivity and specificity (71% and 80% respectively) for the detection of pulmonary TB[21]. However, TB prevalence surveys reveal a high proportion of people diagnosed with pulmonary TB who do not report symptoms[22], and other studies highlight poor implementation and yield of symptom screening[23]. Moreover, many CAD studies have focused on triage of outpatients presenting with symptoms[24–27]. Although there are some examples of CAD screening programs that are not contingent on symptom screening, these have been community-based[28–31].

The aim of this study was to evaluate the diagnostic accuracy of digital CXR with CAD software as a tool for: 1) triage—among patients with cough or TB risk factors—and 2) screening—among patients without cough or TB risk factors—to identify admitted patients who should undergo molecular TB testing in a tertiary care hospital in Lima, Peru.

## Methods

### Study design and participants

We conducted a cross-sectional diagnostic accuracy study that was embedded in a larger prospective study evaluating FAST implementation at Hospital Nacional Hipolito Unanue (HNHU), a 700-bed public, tertiary-care referral hospital in Lima, Peru (https://clinicaltrials.gov/ct2/show/NCT02355223). Patients admitted to HNHU from January 18th 2018 to December 31st 2019 were consecutively screened by the FAST implementation team study staff using a standardized questionnaire upon facility admission, as previously described[11]. This diagnostic accuracy sub-study consisted of two cohorts: triage and screening. Individuals who were eligible for the parent FAST study were eligible for the triage cohort; adults ($\geq$ 18 years old) who, upon questioning by the study team, reported either cough of any duration and/or the following risk factors for TB: contact with someone diagnosed with pulmonary TB, a current active TB diagnosis (however patients who were already on TB treatment were subsequently excluded from this diagnostic accuracy sub-study), or a history of prior active TB. The screening cohort consisted of individuals who were assessed for eligibility for the parent FAST study but were ineligible because they did not have cough or TB risk factors. The rationale for adding a screening cohort to the diagnostic accuracy sub-study was to see the number of patients admitted in our setting in Lima without identified TB risk who may have undiagnosed TB (based on prevalence survey data from other higher TB incidence settings[22]. Every one in five patients with a negative symptom or TB risk screen (undertaken by our FAST implementation study team) was randomly approached for enrollment into the screening cohort for this diagnostic sub-study.

### Ethics statement

The study was approved by the Institutional Review Boards of HNHU and Brigham and Women's Hospital. Written informed consent was obtained from all patients. Participants were assigned a unique study ID number, recorded on data collection forms and clinical specimens to facilitate data linkage; names and other obvious identifiers were not used on data collection forms, thus authors did not have access to information that could identify individual participants during or after data collection.

### Study procedures, data collection, and outcome classification

On the day of admission, patients in both cohorts who were admitted through the emergency room underwent posterior-anterior digital CXR and study staff collected at least 2 sputum samples for TB testing using smear microscopy, mycobacterial culture, Xpert MTB/RIF (Xpert, Cepheid, Sunnyvale, CA), and/or GenoType MTBDRplus line probe assay (Hain, Germany). De-identified CXR images were electronically transferred for automated analysis and were blinded to other demographic and clinical data including the results of other TB testing by the developers of qXR (qure.ai, Mumbai, India) who ran versions 3.0 (v3) and 4.0 (v4) on all images. CXR was obtained prospectively but qXR results were not used to guide clinical management. Information on socio-demographic and clinical variables including current and prior TB history, co-morbidities, and microbiological test results, was collected at the time of enrollment, or retrieved from the medical records using standardized case report forms. Culture and Xpert results were classified separately as binary variables (positive or negative for *Mycobacterium tuberculosis*). If a patient had more than one culture result and at least one was positive, the binary result was classified as positive and the same applied to Xpert results.

### Analyses

For our primary diagnostic accuracy analyses, the diagnosis of pulmonary TB in both the triage and screening cohorts was established by the presence of a sputum culture that grew *Mycobacterium tuberculosis*. For our secondary diagnostic accuracy analyses, the diagnosis of pulmonary TB in both the triage and screening cohorts was established by the presence of a positive sputum Xpert result. Analyses using qXR v4 are presented in the main manuscript and qXR v3 are presented in the supplementary data. qXR sensitivity and specificity (with exact 95% C.I.s) for pulmonary TB were calculated using the manufacturer's prespecified thresholds (0.5 for v3 and v4) per STARD guidelines (see S1 Checklist [32]. DeLong's non-parametric method was applied to compare differences between the areas under the receiver operating characteristic curve (AUC) for the two qXR software versions. We also estimated the specificity at the threshold score at which sensitivity was closest to 90% (WHO triage test minimum TPP recommended criteria[33]. Pre-specified sensitivity analyses were designed to examine qXR accuracy when certain groups known to have increased risk for TB were excluded: people with HIV, people with prior TB, and people with other respiratory diseases (asthma or bronchiectasis).

Using Fisher's exact test, we assessed performance differences in prespecified groups with characteristics or risk factors that may impact diagnostic test performance: male sex, older age, prior TB, HIV co-infection, other respiratory disease co-morbidities, presence of TB symptoms in WHO symptom screen (cough, fever, night sweats, weight loss), and higher-grade sputum smear result. Analyses were completed using STATA/IC version 16 (StataCorp. 2019. Stata Statistical Software: Release 16. College Station, TX: StataCorp LLC.).

## Results

During the study period we enrolled 1006 patients admitted to HNHU who had cough or TB risk factors, of whom 489 underwent digital CXR in the triage cohort (Fig 1). Participants who were taking TB treatment or had been on TB treatment within one year of enrollment (n = 50; 10%) were excluded as were those who had no microbiological testing (n = 20; 4%). We enrolled 220 individuals without cough or TB risk factors in the screening cohort. Screening participants who were household contacts of people who experienced TB were excluded (n = 27; 13%) as were those who had no microbiological testing (n = 9; 4%).

### Triage cohort

**Demographics.**    Of the 419 participants in the triage cohort, 387 (93%) had a mycobacterial culture result that was positive in 65 (17%) participants, of whom 41 (63%) also had positive sputum-smear microscopy results. In this cohort, 398 (95%) had an Xpert MTB/RIF result; it was positive in 69 (17%), of whom 39 (57%) had positive smear microscopy. Culture and Xpert results were largely concordant, with high Xpert sensitivity for both smear-positive and negative culture confirmed TB (95% and 86%), although Xpert was positive in some people who did not have culture or who had a negative culture (S1A and S1B Table). Compared to participants without TB (based on sputum culture results), participants with culture confirmed TB were more likely to be younger, male, have a history of incarceration, report cough longer than 2 weeks, fever, or weight loss, and not have a history of any respiratory diseases or a prior history of TB (Table 1). The primary reason for excluding patients from the triage cohort was that they were not admitted through the emergency department (n = 397/517), which was required for us to be able to obtain dCXR. Differences between included versus excluded patients are described in S2 Table.

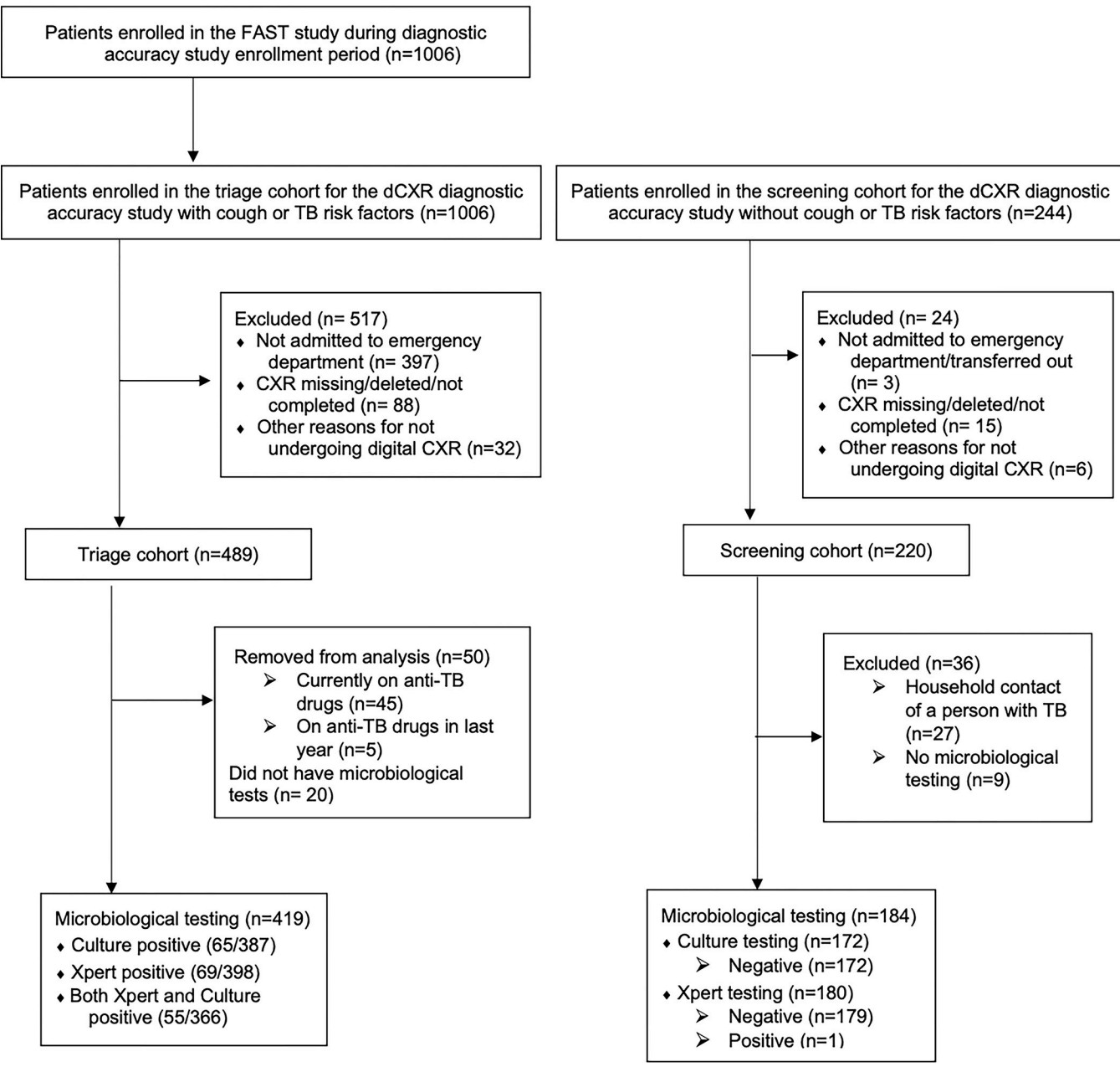

**Fig 1. Study flow diagram.**

## Diagnostic accuracy

Using culture as the reference standard for pulmonary TB, qXR v4 (at the manufacturer pre-specified threshold of 0.5) had an overall sensitivity for pulmonary TB of 0.91 (59/65, 95% CI 0.81–0.97), specificity of 0.32 (103/322, 95% CI 0.2731–0.37), and AUC of 0.78 (95% CI 0.72–0.84) (Table 2). Using Xpert as the reference standard for pulmonary TB, qXR v4 (at the manufacturer pre-specified threshold of 0.5) had an overall sensitivity of 0.93 (64/69, 95% CI 0.84–0.98), specificity of 0.32 (106/329, 95% CI 0.27–0.38), and AUC of 0.76 (95% CI 0.69–0.82) (Table 2). Using a combined reference standard that was positive if either culture or Xpert was

**Table 1. Demographic and clinical characteristics of enrolled participants.**

| | | Triage Patients | | | | Screening* Patients | |
|---|---|---|---|---|---|---|---|
| | Overall (n = 419) | TB^ (n = 65) | No TB (n = 322) | No Culture Performed (n = 32) | P-value** | Overall (n = 184) | P-value** |
| **Median Age (years, interquartile range)** | 41.35 (26.8, 56.6) | 35.34 (24.0, 48.6) | 42.01 (27.5, 57.0) | 44.68 (31.7, 63.3) | **0.003** | 36.19 (25.19, 50.53) | **0.015** |
| **Sex, No (%)** | | | | | | | |
| Female | 164 (39.1) | 17 (26.1) | 134 (41.6) | 13 (40.6) | **0.025** | 111 (60.3) | **<0.001** |
| Male | 255 (60.9) | 48 (73.9) | 188 (58.4) | 19 (59.4) | | 73 (39.7) | |
| **History of Previous TB, No (%)** | | | | | | | |
| Yes | 140 (33.4) | 13 (20.0) | 114 (35.4) | 13 (40.6) | **0.014** | 0 (0.00) | **<0.001** |
| No | 278 (66.4) | 52 (80.0) | 207 (64.3) | 19 (59.4) | | 184 (100) | |
| Refused | 1 (0.20) | 0 (0.00) | 1 (0.3) | 0 (0.00) | | 0 (0.00) | |
| **HIV, No (%)** | | | | | | | |
| Yes | 36 (8.6) | 7 (10.8) | 28 (8.7) | 1 (3.1) | 0.635 | 1 (0.5) | **<0.001** |
| No | 383 (91.4) | 58 (89.2) | 294 (91.3) | 31 (96.9) | | 183 (99.5) | |
| **Smoking, No (%)** | | | | | | | |
| Never | 202 (48.2) | 28 (43.0) | 162 (50.3) | 12 (37.5) | 0.501 | 99 (53.8) | 0.637 |
| Former | 161 (38.4) | 30 (46.2) | 116 (36.0) | 15 (46.9) | | 51 (27.7) | |
| Current | 56 (13.4) | 7 (10.8) | 44 (13.7) | 5 (15.6) | | 34 (18.5) | |
| **Alcohol, No (%)** | | | | | | | |
| Never | 107 (25.5) | 9 (13.9) | 90 (28.0) | 8 (25.0) | 0.092 | 32 (17.3) | **<0.001** |
| Former | 121 (28.9) | 22 (33.9) | 90 (28.0) | 9 (28.1) | | 33 (18.0) | |
| Current | 189 (45.1) | 32 (49.2) | 142 (44.0) | 15 (46.9) | | 119 (64.7) | |
| Missing | 2 (0.5) | 2 (3.0) | 0 (0.00) | 0 (0.00) | | 0 (0.00) | |
| **Respiratory Disease, No (%)** | | | | | | | |
| Asthma | 28 (6.7) | 1 (1.5) | 26 (8.1) | 2 (6.3) | **0.047** | 2 (1.1) | **0.001** |
| Bronchiectasis | 13 (3.1) | 0 (0.00) | 11 (3.4) | 1 (3.1) | | 0 (0.00) | |
| None | 378 (90.2) | 64 (98.5) | 285 (88.5) | 29 (90.6) | | 182 (98.9) | |
| **Diabetes, Type II, No (%)** | | | | | | | |
| Yes | 58 (13.8) | 9 (13.9) | 42 (13.0) | 7 (21.9) | 0.842 | 25 (13.6) | 1.000 |
| No | 361 (86.2) | 56 (86.1) | 280 (87.0) | 25 (78.1) | | 159 (86.4) | |
| **Prison, No (%)** | | | | | | | |
| Yes | 62 (14.8) | 16 (24.6) | 41 (12.7) | 5 (15.6) | **0.020** | 3 (1.6) | **<0.001** |
| No | 357 (85.2) | 49 (75.4) | 281 (87.3) | 27 (84.4) | | 181 (98.4) | |
| **Household Contact of TB positive patient, No (%)** | | | | | | | |
| Yes | 159 (38.0) | 27 (41.5) | 119 (37.0) | 13 (40.6) | 0.754 | - | - |
| No | 256 (61.1) | 38 (58.5) | 199 (61.8) | 19 (59.4) | | | |
| Missing | 4 (0.9) | 0 (0.00) | 4 (1.2) | 0 (0.00) | | | |
| **Smear Status, No (%)** | | | | | | | |
| Positive | 48 (11.5) | 41 (63.1) | 4 (1.2) | 3 (9.4) | **<0.001** | 0 (0.00) | **<0.001** |
| Negative | 363 (86.6) | 16 (24.6) | 318 (98.8) | 29 (90.6) | | 183 (99.5) | |
| Missing | 8 (1.9) | 8 (12.3) | 0 (0.00) | 0 (0.00) | | 1 (0.5) | |
| **TB-associated Symptoms** | | | | | | | |
| **Cough, No (%)** | | | | | | | |
| *Length, in Weeks* | | | | | | | |
| Less than 1 week | 102 (24.4) | 8 (12.3) | 90 (28.0) | 4 (12.5) | **0.003** | - | - |
| 1–2 weeks | 107 (25.5) | 16 (24.6) | 81 (25.1) | 10 (31.2) | | | |
| More than 2 weeks | 189 (45.1) | 39 (60.0) | 132 (41.0) | 18 (56.3) | | | |

*(Continued)*

**Table 1.** (Continued)

| | Overall (n = 419) | Triage Patients | | | | Screening* Patients | |
|---|---|---|---|---|---|---|---|
| | | TB^ (n = 65) | No TB (n = 322) | No Culture Performed (n = 32) | P-value** | Overall (n = 184) | P-value** |
| Missing | 21 (5.0) | 2 (3.1) | 19 (5.9) | 0 (0.00) | | | |
| *Phlegm* | | | | | | | |
| Yes | 352 (84.0) | 60 (92.3) | 262 (81.4) | 30 (93.8) | 0.056 | - | - |
| No | 47 (11.2) | 3 (4.6) | 42 (13.0) | 2 (6.2) | | | |
| Missing | 20 (4.8) | 2 (3.1) | 18 (5.6) | 0 (0.00) | | | |
| *Blood* | | | | | | | |
| Yes | 166 (39.6) | 33 (50.8) | 120 (37.3) | 13 (40.6) | 0.068 | - | - |
| No | 233 (55.6) | 30 (46.2) | 184 (57.1) | 19 (59.4) | | | |
| Missing | 20 (4.8) | 2 (3.0) | 18 (5.6) | 0 (0.00) | | | |
| **Fever, No (%)** | | | | | | | |
| Yes | 265 (63.3) | 52 (80.0) | 192 (59.6) | 21 (65.6) | **0.001** | 85 (46.2) | **<0.001** |
| No | 153 (36.5) | 12 (18.5) | 130 (40.4) | 11 (34.4) | | 99 (53.8) | |
| Refused | 1 (0.2) | 1 (1.5) | 0 (0.00) | 0 (0.00) | | 0 (0) | |
| **Night Sweats in the last 3 months, No (%)** | | | | | | | |
| Yes | 251 (59.9) | 45 (69.2) | 182 (56.5) | 24 (75.0) | 0.072 | 52 (28.3) | **<0.001** |
| No | 168 (40.1) | 20 (30.8) | 140 (43.5) | 8 (25.0) | | 132 (71.7) | |
| **Weight Loss (unintentional), No (%)** | | | | | | | |
| Yes | 293 (69.9) | 55 (84.6) | 218 (67.7) | 20 (62.5) | **0.004** | 84 (45.6) | **<0.001** |
| No | 123 (29.4) | 9 (13.9) | 102 (31.7) | 12 (37.5) | | 98 (53.3) | |
| Refused | 3 (0.7) | 1 (1.5) | 2 (0.6) | 0 (0.00) | | 2 (1.1) | |
| **Difficulty Breathing, No (%)** | | | | | | | |
| Yes | 335 (80.0) | 51 (78.5) | 260 (80.8) | 24 (75.0) | 0.732 | 52 (28.3) | **<0.001** |
| No | 84 (20.0) | 14 (21.5) | 62 (19.2) | 8 (25.0) | | 132 (71.7) | |

^ TB was diagnosed based on positive sputum culture i.e., pulmonary TB, we did not include clinical diagnoses or include evaluation for extra-pulmonary TB

*Screening cohort consists of patients who did not report cough or TB risk factors

** Fisher's exact test on binary variables, chi-square test for categorical variables, Wilcoxon rank sum test for continuous variables, and Jonckeere-Terpstra test for ordered categorical variables. The first p value represents a comparison between participants with and without pulmonary TB in the triage cohort and the second p value represents the comparison between the overall triage and screening cohort participant groups. The missing and refused categories are excluded from statistical comparisons.

positive, sensitivity and specificity for qXR v4 were similar (0.93 and 0.33 respectively) (S3 Table). When the threshold was set such that sensitivity was 90% to match the WHO triage test accuracy performance criterion, specificity was 0.44 (142/322, 95% CI 0.39–0.50) and 0.38 (126/329, 95% CI 0.33–0.44) using the culture and Xpert reference standards respectively (Table 2). Diagnostic accuracy results for qXR v3 are in S1 Text and S4 Table.

There was no difference between the AUCs for qXR v4 and qXR v3 using either the culture reference standard (0.779 [95% CI 0.72–0.84] versus 0.780 [95% CI 0.72–0.84; p = 0.821]) or the Xpert reference standard (0.756 [95% CI 0.69–0.82] versus 0.759 [95% CI 0.70–0.82; p = 0.475) (Fig 2).

## Stratified analyses

There was no difference in qXR v4 sensitivity when stratified by sex, age, prior TB, HIV, and symptoms (Fig 3). qXR v4 sensitivity appeared to be higher in smear-positive compared to smear-negative disease but did not reach statistical significance and numbers of participants

**Table 2. Summary of diagnostic accuracy for qXR version 4 using the culture (primary) and Xpert (secondary) reference standards in the triage and screening cohorts.**

| | Triage Cohort (n = 419) | | | Screening Cohort (n = 184) | | |
|---|---|---|---|---|---|---|
| | Sensitivity (95% CI) | Specificity (95% CI) | AUC (95% CI) | Sensitivity (95% CI) | Specificity (95% CI) | AUC (95% CI) |
| Culture | | | | | | |
| Manufacturer Threshold 0.5 | 90.8% 59/65 (81–96.5%) | 32.0% 103/322 (26.9–37.4%) | 0.779 (0.716, 0.843) | ^ | 93.6% 161/172 (88.8–96.4%) | - |
| Threshold 0.7* | 90.8% 59/65 (81–96.5%) | 44.1% 142/322 (38.6–49.7%) | - | ^ | 96.5% (166/172) (92.4–98.4%) | - |
| Xpert | | | | | | |
| Manufacturer Threshold 0.5 | 92.8% 64/69 (83.9–97.6%) | 32.2% 106/329 (27.2–37.6%) | 0.756 (0.693, 0.819) | 100% 1/1 (2.5–100%) | 93.9% 168/179 (89.3–96.9%) | 0.994 (-, 1.00) |
| Threshold 0.6* | 89.9% 62/69 (80.2–95.8%) | 38.3% 126/329 (33–43.8%) | - | 100% 1/1 (2.5–100%) | 96.6% 173/179 (92.8–98.8%) | - |

*threshold at which sensitivity is closest to 90%

^No positive cultures in the Screening Group

with smear negative disease were low. qXR v4 specificity was higher in people without prior TB than in people with prior TB, with cough less than 2 weeks compared to cough for more than 2 weeks, and with those who did not report weight loss compared to those who reported weight loss (Fig 4). Results for qXR v3 were similar (S1 and S2 Figs).

## Sensitivity analyses

We examined qXR accuracy when pre-specified groups in whom TB diagnostic tests are often less sensitive (PWH, people with prior TB and people with other respiratory diseases) were excluded. Sensitivity for qXR v4 was slightly higher in people without HIV (0.93 [95% CI: 0.83–0.98]), slightly lower in people without prior TB (0.89 [95% CI: 0.77–0.96]), and similar

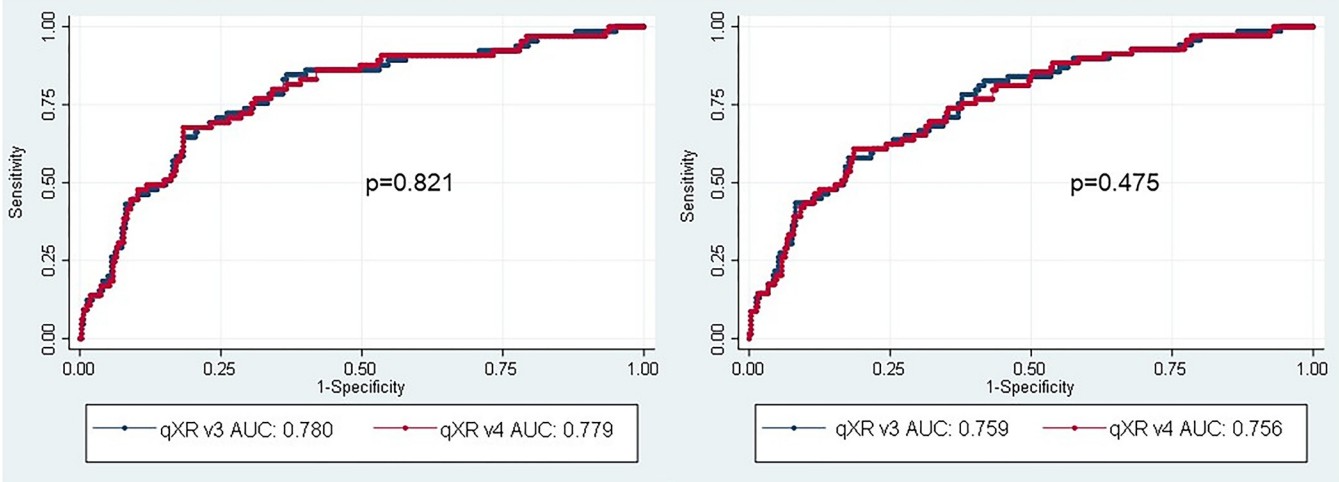

**Fig 2.** Receiver operating characteristic (ROC) curves and estimates of area under the ROC curves (AUC) for qXR versions 3 and 4 to identify abnormalities consistent with TB in the triage cohort using the culture (left) and Xpert (right) reference standards.

| Category and Variable | n/N | p-value | | Sensitivity (95% CI) |
|---|---|---|---|---|
| **Demographics** | | | | |
| Male | 44/48 | 1.000 | | 0.92 (0.80, 0.98) |
| Female | 15/17 | | | 0.88 (0.64, 0.98) |
| <40 years | 35/38 | 1.000 | | 0.92 (0.79, 0.98) |
| >40 years | 24/27 | | | 0.89 (0.71, 0.98) |
| | | | | |
| **Risk Factors** | | | | |
| No Prior TB | 46/52 | 0.828 | | 0.89 (0.77, 0.99) |
| Prior TB | 13/13 | | | 1.00 (0.75, 1.00) |
| HIV Negative | 54/58 | 0.766 | | 0.93 (0.83, 0.98) |
| HIV Positive | 5/7 | | | 0.71 (0.29, 0.96) |
| No Alcohol Use | 8/9 | 1.000 | | 0.89 (0.52, 1.00) |
| Any Alcohol Use | 49/54 | | | 0.91 (0.80, 0.97) |
| Diabetes Negative | 51/56 | 1.000 | | 0.91 (0.80, 0.97) |
| Diabetes Positive | 8/9 | | | 0.89 (0.52, 1.00) |
| | | | | |
| **Clinical Symptoms** | | | | |
| Cough > 2 weeks | 37/39 | 0.850 | | 0.95 (0.83, 0.99) |
| Cough < 2 weeks | 20/24 | | | 0.83 (0.63, 0.95) |
| Night sweats | 39/45 | 0.848 | | 0.87 (0.73, 0.95) |
| No Night sweats | 20/20 | | | 1.00 (0.83, 1.00) |
| Fever | 46/52 | 0.823 | | 0.89 (0.77, 0.96) |
| No Fever | 12/12 | | | 1.00 (0.73, 1.00) |
| Weight Loss | 49/55 | 1.000 | | 0.89 (0.78, 0.96) |
| No Weight Loss | 9/9 | | | 1.00 (0.66, 1.00) |
| | | | | |
| **Test Results** | | | | |
| Smear Positive | 38/41 | 0.830 | | 0.93 (0.80, 0.98) |
| Smear Negative | 13/16 | | | 0.81 (0.54, 0.96) |
| | | | | |
| **Overall** | | | | |
| Overall | 59/65 | | | 0.91 (0.81, 0.96) |

x-axis: 0   .2   .4   .6   .8   1

**Fig 3. Sensitivity of qXR version 4 for culture-confirmed pulmonary tuberculosis, overall and in pre-specified stratified groups.** p values are from Fisher's exact tests.

in people without other respiratory diseases (0.91 [95% CI: 0.81–0.97]). Specificity remained low in people without HIV: 0.31 [95% CI: 0.25–0.36] and people without other respiratory diseases: 0.33 [95% CI: 0.27–0.38], and slightly higher in people without prior TB: 0.40 [95% CI: 0.34–0.47] (S5 Table).

| Category and Variable | n/N | p-value | | Specificity (95% CI) |
|---|---|---|---|---|
| **Demographics** | | | | |
| Male | 49/188 | 0.068 | | 0.26 (0.20, 0.33) |
| Female | 54/134 | | | 0.40 (0.32, 0.49) |
| <40 years | 52/150 | 0.499 | | 0.35 (0.27, 0.43) |
| >40 years | 51/172 | | | 0.30 (0.23, 0.37) |
| | | | | |
| **Risk Factors** | | | | |
| No Prior TB | 84/207 | 0.001 | | 0.41 (0.34, 0.48) |
| Prior TB | 19/114 | | | 0.17 (0.10, 0.25) |
| HIV Negative | 90/294 | 0.252 | | 0.31 (0.25, 0.36) |
| HIV Positive | 13/28 | | | 0.46 (0.28, 0.66) |
| No Alcohol Use | 25/90 | 0.525 | | 0.28 (0.19, 0.38) |
| Any Alcohol Use | 78/232 | | | 0.34 (0.28, 0.40) |
| Diabetes Negative | 88/280 | 0.740 | | 0.31 (0.26, 0.37) |
| Diabetes Positive | 15/42 | | | 0.36 (0.22, 0.52) |
| | | | | |
| **Clinical Symptoms** | | | | |
| Cough > 2 weeks | 24/132 | 0.001 | | 0.18 (0.12, 0.26) |
| Cough < 2 weeks | 73/171 | | | 0.43 (0.35, 0.51) |
| Night sweats | 56/182 | 0.733 | | 0.31 (0.24, 0.38) |
| No Night sweats | 47/140 | | | 0.34 (0.26, 0.42) |
| Fever | 57/192 | 0.491 | | 0.30 (0.23, 0.37) |
| No Fever | 46/130 | | | 0.35 (0.27, 0.44) |
| Weight Loss | 57/218 | 0.024 | | 0.26 (0.20, 0.33) |
| No Weight Loss | 46/102 | | | 0.45 (0.35, 0.55) |
| | | | | |
| **Overall** | | | | |
| Overall | 103/322 | | | 0.32 (0.27, 0.38) |

**Fig 4. Specificity of qXR version 4 for culture-confirmed pulmonary tuberculosis, overall and in pre-specified stratified groups.** p values are from Fisher's exact tests.

## High prevalence of lung abnormalities

A high prevalence of radiographic lung abnormalities, most notably opacities (81%), consolidation (62%), fibrosis (47%), nodules (58%), or cavitation (19%), was detected by qXR on digital CXR images from the triage cohort (S6 Table).

### Screening cohort

Compared to participants in the triage cohort, participants in the screening cohort were more likely to be younger and female, not have a history of HIV, any respiratory diseases or a prior history of TB, not have a history of incarceration, more likely to report current alcohol use, and less likely to report fever, night sweats, or weight loss (Table 1). No participants in the screening cohort had a positive culture, and only one participant had a positive Xpert. Since there was only one person with confirmed TB in the screening group (who did have a qXR positive result), we only report specificity. Using the manufacturer's pre-specified thresholds, the specificity for qXR v4 was 0.94 (95% CI 0.89–0.96) using the culture reference standard and 0.94 (95% CI 0.89–0.97) using the Xpert reference standard (Table 2).

## Discussion

In our study population of hospitalized patients at a tertiary referral hospital in Lima, Peru, the use of qXR artificial intelligence software analysis versions 3 and 4 in a triage cohort of patients with cough or TB risk factors demonstrated a high sensitivity (>90%) but low specificity (~30%), thereby meeting only the WHO triage test criteria for sensitivity. In our screening cohort of patients without cough or risk factors, specificity was high (>90%) but sensitivity could not be evaluated since the diagnostic yield of screening this group in this setting was low (only one patient was diagnosed with Xpert-positive TB).

We previously reported that the FAST strategy using Xpert for molecular diagnosis increased the yield of TB diagnosis and decreased time to treatment initiation[11]. Yet, despite WHO guidance that molecular WHO-recommended rapid TB diagnostic tests (mWRD) such as Xpert should be the initial test for people being evaluated for TB, implementation in Peru and other high-incidence settings has lagged[1]. While barriers to mWRD implementation are multifactorial[34], cost and limited laboratory capacity were challenges to the implementation of Xpert as a triage or screening test as part of routine practice in our setting. The use of a triage tool such as digital CXR with CAD can help identify which patients should undergo testing with a mWRD[16] as part of transmission prevention strategies such as FAST. In our hospitalized study population, qXR was highly sensitive for correctly triaging people identified as having cough or TB risk factors who had culture confirmed disease. Although low qXR specificity would lead to a large number of patients with false positive results who required confirmatory testing and widespread use of digital CXR with CAD poses implementation challenges, qXR as a triage tool could be of clinical and public health value due to its impact on diagnostic yield and may still save enough mWRDs to be cost-effective depending on the setting (cost-effectiveness analyses from our study are forthcoming). When we adjusted the threshold for qXR v4 to maintain sensitivity at 90%, specificity rose to 38–44%; thus our data add further weight to the need for population-specific thresholds[35] to optimize implementation of CAD tools in different settings.

The low specificity of qXR in inpatients with TB symptoms or risk factors contrasts with cross-sectional studies that found that qXR met WHO triage test criteria for both sensitivity (>90%) and specificity (70%) when evaluated in symptomatic outpatients in Bangladesh and Pakistan[24, 36]. Our triage cohort had a high prevalence of radiographic lung abnormalities, which was likely to be an important contributing factor to the lower than expected specificity in this cohort. Abnormal chest imaging findings in our study population may be due to inpatient populations in a tertiary referral hospital being more likely to have acute illnesses such as pneumonia, and may also reflect a higher proportion of people with chronic lung disease in Lima, a city known to have high rates of air pollution, which has also been associated with a higher risk of tuberculosis[37]. We also note that this diagnostic accuracy assessment in the

triage cohort reflects use of the test in a pre-screened population who had a high pre-test probability of TB or other lung disease and underwent microbiological testing that revealed a high prevalence of TB. Thus, negative predictive value would be lower for this cohort than if qXR testing was applied to the population of people initially screened (rather than those enrolled) for FAST.

Increasing data demonstrate symptom screening is insensitive[38] and often poorly implemented[23], and a high proportion of people with TB do not report symptoms[22]. The inclusion of individuals without cough or risk factors in our screening cohort was designed to try to understand the potential diagnostic yield of using qXR as a screening tool to identify unsuspected TB in hospitalized patients who may be presenting for various other reasons. In this setting, the diagnostic yield of screening people without symptoms or risk factors was lower than expected (based on outpatient studies). The specificity of qXR was high, suggesting it could be a valuable rule-out test in this setting. The low prevalence of TB in the screening cohort may be an artifact of the sample size or, it may be because people with TB who present to hospital are more likely to be sicker due to TB and thus present with cough (resulting in exclusion from the screening cohort) compared to the outpatient populations in prevalence surveys. The exclusion of people with TB contacts and prior TB from the screening cohort may have also led to the screening cohort being a lower risk group. The implementation of strategies such as FAST should consider local epidemiology—including the pre-test probability of TB in people who do not report symptoms—to determine the optimal approach to determining who should undergo mWRD testing. Other strategies could also be evaluated to increase the sensitivity of screening.

Strengths of our study include generating CAD diagnostic accuracy data from inpatient populations, including those who were symptomatic and/or high-risk and those without identified cough or TB risk factors, also contributing to a body of literature seeking to optimize the FAST facility-based transmission prevention strategy in a medium incidence setting. We provide the first head-to-head evaluation of version 4 (soon to be commercially available) compared to qXR version 3 and characterize other lung abnormalities detected. We acknowledge the challenges posed by imperfect reference standards for TB diagnostic accuracy studies[16], although we suspect that paucibacillary disease (which could cause culture, Xpert, and also CXR to be negative) is less likely in a hospitalized cohort in a low-HIV prevalence setting. Moreover, the inclusion of reference standard data from both mycobacterial culture and Xpert is a strength since many diagnostic studies only use Xpert as the refence standard. Limitations of our study are that digital CXR could only be performed on inpatients admitted through the emergency room (which may bias the study towards sicker hospitalized patients) and that with only 65 patients who had culture-confirmed TB, the study only had sufficient power such that we can report the lower limit of the 95% CI for sensitivity is 0.885 with 95% precision. We note low numbers in certain subgroups, including the number with HIV due to the low incidence of HIV in Peru and number with smear negative disease, also limit the power to detect differences in our stratified analyses.

In conclusion, qXR had high sensitivity but low specificity as a triage tool in the context of use within the FAST strategy in hospitalized adults admitted to a tertiary referral hospital in Peru who had a high prevalence of other radiographic lung abnormalities. While specificity was high in patients without cough or risk factors, the diagnostic yield of screening these patients was low in this setting. These findings further support the need for population and setting-specific thresholds for CAD programs and provide additional insights into the role for triage testing in hospitalized patients, which remains critical to detect and treat individual patients earlier and to curb hospital TB transmission.

## Supporting information

**S1 Checklist. Reporting checklist for diagnostic test accuracy studies.**
(DOCX)

**S1 Table.** a: Summary of Culture and Xpert results concordance. b: Xpert sensitivity for smear-positive and smear-negative culture-positive TB in triage cohort patients.
(DOCX)

**S2 Table. Demographic and clinical characteristics of enrolled participants compared to those who were excluded.**
(DOCX)

**S3 Table. Diagnostic accuracy of qXR Version 3 and 4 using a reference standard which is positive if either mycobacterial culture or Xpert is positive for the triage cohort.**
(DOCX)

**S4 Table. Summary of diagnostic accuracy for qXR version 3 compared to the culture (primary) and Xpert (secondary) reference standards.**
(DOCX)

**S5 Table. Diagnostic accuracy of qXR Version 3 and 4 for pre-specified subgroups in the triage cohort for which participants with prior TB, respiratory diseases, and HIV, were excluded.**
(DOCX)

**S6 Table. Lung abnormalities detected by qXR analysis for the triage and screening cohorts.**
(DOCX)

**S1 Fig. Sensitivity of qXR version 3 for culture-confirmed pulmonary tuberculosis, overall and in prespecified stratified groups.** p values are from Fisher's exact tests.
(TIFF)

**S2 Fig. Specificity of qXR version 3 for culture-confirmed pulmonary tuberculosis, overall and in prespecified stratified groups.** p values are from Fisher's exact tests.
(TIFF)

**S1 Text.**
(DOCX)

## Acknowledgments

The authors were allowed to use the qXR algorithms free of charge from qure.ai for research purposes, but the companies had no influence over the research question, nor any other aspect of the work carried out, and had no impact on the transparency of the article.

## Author Contributions

**Conceptualization:** Carole D. Mitnick, Edward Nardell, Dylan B. Tierney, Ruvandhi R. Nathavitharana.

**Data curation:** Amanda M. Biewer, Christine Tzelios, Karen Tintaya, Betsabe Roman, Leonid Lecca, Ruvandhi R. Nathavitharana.

**Formal analysis:** Amanda M. Biewer, Christine Tzelios, Ruvandhi R. Nathavitharana.

**Funding acquisition:** Edward Nardell, Dylan B. Tierney, Ruvandhi R. Nathavitharana.

**Investigation:** Karen Tintaya, Betsabe Roman, Edward Nardell, Leonid Lecca, Dylan B. Tierney, Ruvandhi R. Nathavitharana.

**Methodology:** Karen Tintaya, Betsabe Roman, Shelley Hurwitz, Courtney M. Yuen, Edward Nardell, Leonid Lecca, Dylan B. Tierney, Ruvandhi R. Nathavitharana.

**Project administration:** Karen Tintaya, Betsabe Roman, Edward Nardell, Leonid Lecca, Dylan B. Tierney, Ruvandhi R. Nathavitharana.

**Resources:** Edward Nardell, Dylan B. Tierney, Ruvandhi R. Nathavitharana.

**Software:** Ruvandhi R. Nathavitharana.

**Supervision:** Karen Tintaya, Betsabe Roman, Edward Nardell, Leonid Lecca, Dylan B. Tierney, Ruvandhi R. Nathavitharana.

**Validation:** Amanda M. Biewer, Christine Tzelios, Karen Tintaya, Betsabe Roman, Shelley Hurwitz, Courtney M. Yuen, Carole D. Mitnick, Edward Nardell, Leonid Lecca, Dylan B. Tierney, Ruvandhi R. Nathavitharana.

**Visualization:** Amanda M. Biewer, Christine Tzelios, Shelley Hurwitz, Courtney M. Yuen, Carole D. Mitnick, Edward Nardell, Dylan B. Tierney, Ruvandhi R. Nathavitharana.

**Writing – original draft:** Amanda M. Biewer, Christine Tzelios, Ruvandhi R. Nathavitharana.

**Writing – review & editing:** Karen Tintaya, Betsabe Roman, Shelley Hurwitz, Courtney M. Yuen, Carole D. Mitnick, Edward Nardell, Leonid Lecca, Dylan B. Tierney, Ruvandhi R. Nathavitharana.

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
