## [Decision Letter · Decision Letter 0]

21 Aug 2023

PGPH-D-23-00918

Accuracy of digital chest x-ray analysis with artificial intelligence software as a triage and screening tool in hospitalized patients being evaluated for tuberculosis in Lima, Peru.

Dear Dr. Nathavitharana,

Thank you for submitting your manuscript to PLOS Global Public Health. After careful consideration, we feel that it has merit but does not fully meet PLOS Global Public Health’s publication criteria as it currently stands. Therefore, we invite you to submit a revised version of the manuscript that addresses the points raised during the review process.

We look forward to receiving your revised manuscript.

Kind regards,

Andrew D. Kerkhoff

Academic Editor

Journal Requirements:

2. We noticed that you used "unpublished" in the manuscript. We do not allow these references, as the PLOS data access policy requires that all data be either published with the manuscript or made available in a publicly accessible database. Please amend the supplementary material to include the referenced data or remove the references.

3. We notice that your supplementary figures are uploaded with the file type 'Figure'. Please amend the file type to 'Supporting Information'. Please ensure that each Supporting Information file has a legend listed in the manuscript after the references list.

Additional Editor Comments (if provided):

Reviewers' comments:

Reviewer's Responses to Questions

**Comments to the Author**

1. Does this manuscript meet PLOS Global Public Health’s publication criteria? Is the manuscript technically sound, and do the data support the conclusions? The manuscript must describe methodologically and ethically rigorous research with conclusions that are appropriately drawn based on the data presented.

Reviewer #1: Yes

Reviewer #2: Yes

2. Has the statistical analysis been performed appropriately and rigorously?

Reviewer #1: Yes

Reviewer #2: Yes

3. Have the authors made all data underlying the findings in their manuscript fully available (please refer to the Data Availability Statement at the start of the manuscript PDF file)?

Reviewer #1: Yes

Reviewer #2: Yes

4. Is the manuscript presented in an intelligible fashion and written in standard English?

Reviewer #1: Yes

Reviewer #2: Yes

5. Review Comments to the Author

Reviewer #1: Very well written paper, with an overall sound analysis. Provides needed data on a novel application of cad technology: assessment of people being hospitalized to reduce nosocomial transmission of TB.

I recommend the authors address the following to facilitate reader evaluation of the study, and to strengthen some aspects of the reporting and interpretation to be more consistent with the methods and results:

- This was a cross-sectional study embedded in a larger prospective study. Please add a participant flow diagram that relates the sub-study population to the larger study population of the prospective study.

- Please describe how patients were selected for enrollment into the substudy. Randomly approached, vs consecutively etc

- It seems inconsistent that a risk factor that permitted someone to be eligible was 'current active TB diagnosis' but then people on active TB treatment were excluded. Can you please edit the phrasing so that it is no longer inconsistent?

- If only 2 sputum samples were submitted, 1 for culture and 1 for Xpert, how could some participants have more than one culture result or more than one Xpert (lines 130-132)?

- A single sputum culture is an imperfect reference standard for pulmonary TB. A single sputum culture is about 73% sensitive for smear-positive TB, and only 61% sensitive for smear-negative TB (see Nelson et al. 1998). A combination of two cultures is over 90% sensitive for smear-negative TB. Similar limitations apply to Xpert. As such, a useful sensitivity analysis would be to assess qxr sensitivity and specificity against a reference standard that includes both TB culture and Xpert (where if either one is positive then the participant is classified as having active TB).

- Can the authors comment on above in the discussion: do they consider possible source of bias or not?

- What was rationale for excluding household contacts from the screening participant cohort? Why not include them in the triage cohort? (line 162-163)

- could the authors comment on differences in characteristics between included and excluded participants-- in Results, and perhaps add a supplement table?

- could the authors report Xpert sensitivity for smear-negative culture-positive TB in this cohort?

- was there a reason that alcohol use disorder and diabetes are reported but not evaluated as stratification variables?

- I have major concerns about the interpretation of the stratified analyses for HIV status.

The sample size was small, and likely the stratification was underpowered to detect a difference. The point estimate for sensitivity of qxr for PLHIV is 27% lower. I do not agree with the authors' interpretation that HIV had no impact on sensitivity.

Similary for specificity, sample size remains an issue. It is surprising that HIV status was associatd with increased specificity- can the authors compare their result to other literature in the discussion?

Also can the authors clarify why the point estimate of 0.54 is not equal to 13/22?

- For smear-negative disease, point estimates are also lower for sensitivity compared to smear positive diseae (by 10%). Could this difference have been underestimated due to use of single culture or xpert as reference standard? It is biologically plausible that radiograph will be less sensitive for less extensive disease. This would be something of value to comment on in the discussion, and compare to other studies.

- The authors comment on limited precision of their main analysis due to sample size limitations. Would this not also mean some limitations to power to detect differences in stratified analyses? If so- consider adding also as a limitation.

- The use of liquid culture and Xpert is a strength of the study, most studies in the field only use Xpert as the reference standard. I would recommend the authors add this as a strength.

Reviewer #2: The authors have clearly described the methods and results of CAD software evaluation, which shows the software has high sensitivity for TB screening in certain use cases. They have also adequately described the limitations of the sample and yield for the interpretation of software performance in other use cases.

Methods:

The recruitment cascade and participant sample included in the final analysis could be better described. For the triage data set, it is clear – people were enrolled and tested based on symptoms and the majority got a CXR. But it is not clear what the source population is for the screening cohort, nor how they were indicated for TB testing. Was this a subset of the population verbally screened, who had a CXR abnormal result from a radiologist? If this is the case, it would be nice to see the population screened and how it gets split into Cohort 1 & 2 (and their relative sizes). Does the analysis data set include individuals who have CXR normal result from a human reader (live interpretation) and an abnormal result from the CAD software (retrospective interpretation)?

Results/Discussion:

The low yield in the screening cohort is interesting (NNT almost 200). We need to understand how this population was recruited in order to better understand their risk (see comment above). The discussion notes that contacts and people with past history of TB are excluded, maybe making this population lower-risk. But if they were primarily symptom negative, CXR abnormal (by a radiologist), it is also possible that the radiologist has issues with CXR interpretation quality.

The conclusion of low specificity is accurate, but the discussion (and final conclusions paragraph) doesn’t sufficiently put this finding into context. The low specificity calculation is among study participants who are heavily pre-screened and have a microbiological result, not among hospital clients who are screened via FAST. In the context of FAST, True Negative people are likely to be correctly triaged by the symptoms and CXR screens and to never get the opportunity to receive a microbiological test to confirm their True Negative status. Normal CXR images are easier to discerning whether an abnormality is TB related or not. So the evaluation’s low specificity finding is very likely to be underestimating the CAD software’s performance among the entire screened population. If there was a composite reference standard where CXR normal FAST participants were considered a True Negative – or even better a double CXR normal result – you’d likely see significantly higher specificity. Text acknowledging this in the discussion – or at least that this population as not included in your specificity calculations – should be added to the discussion.

6. PLOS authors have the option to publish the peer review history of their article (what does this mean?). If published, this will include your full peer review and any attached files.

**Do you want your identity to be public for this peer review?** For information about this choice, including consent withdrawal, please see our Privacy Policy.

Reviewer #1: No

Reviewer #2: No

---

## [Decision Letter · Decision Letter 1]

28 Nov 2023

PGPH-D-23-00918R1

Accuracy of digital chest x-ray analysis with artificial intelligence software as a triage and screening tool in hospitalized patients being evaluated for tuberculosis in Lima, Peru.

Dear Dr. Nathavitharana,

Thank you for submitting your manuscript to PLOS Global Public Health. After careful consideration, we feel that it has merit but does not fully meet PLOS Global Public Health’s publication criteria as it currently stands. Therefore, we invite you to submit a revised version of the manuscript that addresses the points raised during the review process.

We look forward to receiving your revised manuscript.

Kind regards,

Devan Jaganath

Academic Editor

Journal Requirements:

Additional Editor Comments (if provided):

The comments from the review have overall been addressed; the first reviewer has made an additional comment related to the abstract. I would consider their suggestion, or removing the sentence or clarifying the sentence so that the stratified results are described but not compared given that it is underpowered.

Reviewers' comments:

Reviewer's Responses to Questions

**Comments to the Author**

1. If the authors have adequately addressed your comments raised in a previous round of review and you feel that this manuscript is now acceptable for publication, you may indicate that here to bypass the “Comments to the Author” section, enter your conflict of interest statement in the “Confidential to Editor” section, and submit your "Accept" recommendation.

Reviewer #1: All comments have been addressed

Reviewer #2: All comments have been addressed

2. Does this manuscript meet PLOS Global Public Health’s publication criteria? Is the manuscript technically sound, and do the data support the conclusions? The manuscript must describe methodologically and ethically rigorous research with conclusions that are appropriately drawn based on the data presented.

Reviewer #1: Yes

Reviewer #2: Yes

3. Has the statistical analysis been performed appropriately and rigorously?

Reviewer #1: Yes

Reviewer #2: Yes

4. Have the authors made all data underlying the findings in their manuscript fully available (please refer to the Data Availability Statement at the start of the manuscript PDF file)?

Reviewer #1: Yes

Reviewer #2: Yes

5. Is the manuscript presented in an intelligible fashion and written in standard English?

Reviewer #1: Yes

Reviewer #2: Yes

6. Review Comments to the Author

Reviewer #1: The authors have adequately addressed my concerns. However, I strongly suggest a change to the abstract which would be in keeping with the authors' own comments about power to detect differences in stratified analyses-- which was low. The following cannot be concluded due to the low power: "qXR sensitivity did not differ stratified by sex, age, prior TB, HIV, and symptoms."

Please remove that sentence and replace with a statement that the study was underpowered to detect meaningful differences in sensitivity stratified by sex, age, prior TB, smear status, HIV status, and symptoms.

Minor comment:

The sentence is difficult to follow- can it be clarified? "While dCXR/CAD specificity is typically lower in PWH, in contrast to our findings, we

348 note low numbers in certain subgroups, including the number with HIV due to the low incidence of

349 HIV in Peru and number with smear negative disease, also limit the power to detect differences in our

350 stratified analyses."

Reviewer #2: All comments have been fully addressed.

7. PLOS authors have the option to publish the peer review history of their article (what does this mean?). If published, this will include your full peer review and any attached files.

**Do you want your identity to be public for this peer review?** For information about this choice, including consent withdrawal, please see our Privacy Policy.

Reviewer #1: No

Reviewer #2: No

---

## [Decision Letter · Decision Letter 2]

16 Jan 2024

Accuracy of digital chest x-ray analysis with artificial intelligence software as a triage and screening tool in hospitalized patients being evaluated for tuberculosis in Lima, Peru.

PGPH-D-23-00918R2

Dear Dr. Natahvitharana,

We are pleased to inform you that your manuscript 'Accuracy of digital chest x-ray analysis with artificial intelligence software as a triage and screening tool in hospitalized patients being evaluated for tuberculosis in Lima, Peru.' has been provisionally accepted for publication in PLOS Global Public Health.

Best regards,

Priya Rajendran, PhD

Academic Editor

Reviewer Comments (if any, and for reference):

Reviewer's Responses to Questions

**Comments to the Author**

1. If the authors have adequately addressed your comments raised in a previous round of review and you feel that this manuscript is now acceptable for publication, you may indicate that here to bypass the “Comments to the Author” section, enter your conflict of interest statement in the “Confidential to Editor” section, and submit your "Accept" recommendation.

Reviewer #1: All comments have been addressed

Reviewer #2: All comments have been addressed

2. Does this manuscript meet PLOS Global Public Health’s publication criteria? Is the manuscript technically sound, and do the data support the conclusions? The manuscript must describe methodologically and ethically rigorous research with conclusions that are appropriately drawn based on the data presented.

Reviewer #1: Yes

Reviewer #2: Yes

3. Has the statistical analysis been performed appropriately and rigorously?

Reviewer #1: Yes

Reviewer #2: Yes

4. Have the authors made all data underlying the findings in their manuscript fully available (please refer to the Data Availability Statement at the start of the manuscript PDF file)?

Reviewer #1: Yes

Reviewer #2: Yes

5. Is the manuscript presented in an intelligible fashion and written in standard English?

Reviewer #1: Yes

Reviewer #2: Yes

6. Review Comments to the Author

Reviewer #1: (No Response)

Reviewer #2: All comments have been addressed.

7. PLOS authors have the option to publish the peer review history of their article (what does this mean?). If published, this will include your full peer review and any attached files.

**Do you want your identity to be public for this peer review?** For information about this choice, including consent withdrawal, please see our Privacy Policy.

Reviewer #1: No

Reviewer #2: No
